# Independent Evaluation of Cell Culture Systems for Hepatitis E Virus

**DOI:** 10.3390/v14061254

**Published:** 2022-06-09

**Authors:** Nicholas Chew, Jianwen Situ, Shusheng Wu, Weiming Yao, Siddharth Sridhar

**Affiliations:** 1Department of Microbiology, School of Clinical Medicine, Li Ka Shing Faculty of Medicine, The University of Hong Kong, Hong Kong, China; chewnfs11@connect.hku.hk (N.C.); situjw@hku.hk (J.S.); shushengwu2012@gmail.com (S.W.); yao719082506@126.com (W.Y.); 2State Key Laboratory of Emerging Infectious Diseases, The University of Hong Kong, Hong Kong, China; 3Carol Yu Centre for Infection, The University of Hong Kong, Hong Kong, China

**Keywords:** hepatitis E virus, cell culture, wild-type HEV growth, rat HEV

## Abstract

Hepatitis E virus (HEV) infection in humans is primarily caused by genotypes within *Paslahepevirus* species *balayani* (HEV-A). *Rocahepevirus* species *ratti* (HEV-C1, otherwise known as rat HEV) can also infect humans. HEV grows poorly in cell culture. Recent studies have reported that hyper-confluent cell layers, amphotericin B, MgCl_2_, progesterone, and dimethyl sulfoxide (DMSO) increase HEV yield in vitro. Here, we describe an independent evaluation of the effectiveness of these modifications in improving the yield of HEV-A genotype 4 (HEV-A4) and HEV-C1 from clinical samples in PLC/PRF/5 cells. We found that amphotericin B, MgCl_2_, and DMSO increased HEV yield from high-viral-load patient stool samples, while progesterone was not effective. Yield of HEV-C1 was lower than HEV-A4 across all medium conditions, but was boosted by DMSO. HEV-A4 could be maintained for over 18 months in amphotericin B- and MgCl_2_-containing medium, with the demonstration of viral antigen in supernatants and infected cells. We also evaluated various protocols to remove pseudo-envelopes from cell culture-derived HEV. Treating cell culture supernatant with NP-40 was the most effective. Our findings identify key modifications that boost HEV growth in vitro and illustrate the importance of independent verification of such studies using diverse HEV variants and cell lines.

## 1. Introduction

Hepatitis E virus (HEV) is a significant cause of viral hepatitis in humans, with disease manifestations including asymptomatic infection, acute self-limiting hepatitis, persistent hepatitis, and extrahepatic manifestations [1]. HEV is a positive-sense single-stranded RNA virus; its genome size is approximately 7.2 kB and comprises three open reading frames (ORFs) encoding structural and non-structural proteins [2]. 

The family *Hepeviridae* comprises two subfamilies: *Orthohepevirinae* and *Parahepevirinae*. The former comprises four genera infecting a variety of terrestrial and arboreal animals. *Paslahepevirus balayani* comprises eight genotypes (A1–A8), three of which cause the most hepatitis E in humans (A1, A3, and A4) [3]. HEV-A1 circulates in humans while HEV-A3 and HEV-A4 are transmitted from swine to humans via undercooked pork [4]. The three other genera, *Avihepevirus*, *Rocahepevirus*, and *Chirohepevirus,* predominantly circulate in birds, rodents, and bats, respectively. *Rocahepevirus* species *ratti* (HEV-C1) was first discovered in rats and was considered to have minimal zoonotic potential due to high genetic and antigenic divergence from HEV-A [5,6]. However, we and others have definitively demonstrated that HEV-C1 can infect humans [7,8,9]. As HEV-C1 is not specifically tested for in most parts of the world, the global burden of HEV-C1 is not known. However, we found that it accounted for 8–15% of all virologically confirmed hepatitis E in Hong Kong [10,11].

HEV is difficult to model in cell culture, which has hindered study of this virus [12]. Even when isolation from clinical specimens is possible, the presence of quasi-enveloped HEV (eHEV) in cell culture-derived HEV particles significantly decreases infectivity compared to the native non-enveloped form of the virus (nHEV) [13]. Moderate success has been achieved using infectious clones, such as Kernow-C1, that are highly adapted to cell culture. However, there is a question of whether they truly reflect the behavior of all wild-type HEV isolates, due to the frequent presence of non-viral host-derived inserts (such as the 58 amino acid human S17 ribosomal protein fragment in the hypervariable region of ORF1 of Kernow-C1) that are typically found in some, but not all, wild-type strains isolated from immunocompromised patients [14,15,16]. This emphasizes the need for an effective and reproducible cell culture model for wild-type HEV, enabling the study of the HEV replication cycle, as well as the design of serum neutralization tests and the identification of promising antiviral candidates. 

Three protocols have recently been proposed to optimize the yield of HEV in cell culture [17,18,19]. Key factors identified in these studies that improve culture yield include hyper-confluent cell layers, the addition of dimethyl sulfoxide (DMSO), amphotericin B, MgCl_2_, and progesterone. However, each protocol uses distinct cell lines, basal media, and HEV strains; the sheer number of variables makes it difficult to ascertain whether these findings are generalizable to other variants of HEV. We hypothesized that key cell culture modifications identified by these authors should promote the growth of HEV irrespective of the virus variant or cell line. In this replication study, we tested the effect of combinations of these key cell culture medium modifications on the growth of patient-derived strains of HEV-A4 and HEV-C1in the PLC/PRF/5 human hepatoma cell line. Lastly, we compared the efficacy of various treatments to convert eHEV to nHEV because of its relevance in optimizing HEV cell culture yield given the reduced infectivity of the former.

## 2. Materials and Methods

### 2.1. Cell Lines and Media

The human hepatoma cell line PLC/PRF/5 (ATCC CRL-8024, LGC Standards, Wesel, Germany) was used due to its documented ability to support the growth of both HEV-A and HEV-C1 [19,20,21,22]. A variety of media were prepared, as noted in Table 1. Amphotericin B, magnesium chloride (MgCl_2_), and progesterone were purchased from Sigma-Aldrich (St. Louis, MO, USA); FBS from Thermo Fisher Scientific (Waltham, WA, USA); and DMSO from ATCC (Wesel, Germany). Minimum essential medium (MEM) and phosphate-buffered saline (PBS) were prepared in-house.

### 2.2. HEV Isolates

Fecal samples were derived from two solid organ transplant recipients with chronic hepatitis E infection before ribavirin initiation [11,23]. One patient was infected with HEV-A4 and the other with HEV-C1. Samples were stored at −80 °C before the preparation of inocula. For inoculation, fecal specimens were diluted 1:10 in PBS, thoroughly vortexed, and centrifuged for 10 min at 8000× *g*, and the supernatant was passed through a 0.22 μm sterile filter (Merck, Darmstadt, Germany). The final inoculum had an approximate concentration of 10^5^ copies/mL. Both patients provided written informed consent. The study was approved by the Institutional Review Board of the University of Hong Kong/Hospital Authority West Cluster (UW 18-074).

### 2.3. Propagation of HEV Strains in Cell Culture

At 80–100% confluence, naïve cells were detached using EDTA trypsin (Thermo Fisher Scientific), diluted in the specified media (Table 1) and seeded into a flask or plate at maximum cell density (0.7 × 10^6^ cells for a T-25 flask, 0.3 × 10^6^ cells per well in a 6-well plate). Cells were incubated at 37 °C and 5% CO_2_ for 7–14 days with media being completely refreshed twice weekly until they reached hyper-confluence. On the day of infection, the media were replaced by an equal volume of inoculum, which corresponds to an MOI of approximately 0.1 genome equivalents per cell, and incubated for 75 min at room temperature. The inoculum was then removed and replaced with the specified media (Table 1), followed by incubation at 37 °C and 5% CO_2_. After 24 h, the supernatant was completely refreshed, and 50% of the supernatant was harvested and replaced with fresh media every 3–4 days.

### 2.4. HEV RNA Quantification

Total nucleic acid (TNA) was extracted from 200 µL of plasma or stool filtrate into 60 µL of eluate using the EZ1 Virus Mini Kit v2.0 (Qiagen, Hilden, Germany), or from 200 µL of cell culture supernatant into 30 µL of eluate using the MiniBEST Viral RNA/DNA Extraction Kit v5.0 (Takara Bio, Kusatsu, Japan). TNA was tested by quantitative real-time reverse transcription–polymerase chain reaction (RT-qPCR) for HEV-A or HEV-C1 detection, as described in our previous work [9]. The limit of detection was calculated for our detection assays using the Probit method and was 50 and 160 copies/mL for HEV-A4 and -C1, respectively.

### 2.5. Immunofluorescence Assay

PLC/PRF/5 cells were seeded into a 24-well plate containing a coverslip at a density of 5 × 10^4^ cells per well. Once cells were confluent, the supernatant was removed and 500 µL of inocula was added. The plate was incubated for 75 min at room temperature, after which the inocula was removed and the well was gently washed with 500 µL PBS. It was then re-suspended in 500 µL of MEM-A. After 3 dpi, 250 µL was harvested and replaced with fresh MEM-A. All supernatant was harvested at 7 dpi. Samples were then fixed in 4% (*v*/*v*) paraformaldehyde at room temperature for 60 min and blocked in 1% bovine serum albumin (BSA) containing 0.1% Tween 20 at room temperature for 60 min. The BSA was then removed, and the fixed cells were incubated with rabbit anti-HEV ORF2 polyclonal antibody (HEV-A polyclonal antibody (1:400); HEV-C polyclonal antibody (1:400)) (both were prepared in-house by injection of HEV-A4 p239 and HEV-C1 p241 virus-like particles, as described previously [6] at room temperature for 60 min. After washing with TPBS (0.3% Tween 20 in PBS), the cells were stained with Alexa Fluor 488-conjugated anti-rabbit IgG (1:4000) (A-11008, Thermo Fisher Scientific) at room temperature for 35 min. They were then washed with TPBS 6 times. Nuclei were then counterstained with 4,6-diamidino-2-phenylindole (DAPI).

### 2.6. Detection of HEV ORF2 Antigen

A commercial hepatitis E antigen ELISA kit (Wantai, Beijing, China) was used according to the manufacturer’s instructions for the detection of HEV-A ORF2 antigen in culture supernatant.

### 2.7. Conversion of eHEV to nHEV

Supernatants of PLC/PRF/5 cells infected with HEV-A4 were combined into an 80 mL stock. The stock was ultracentrifuged for 16 h at 160,000× *g*. The precipitate was re-suspended in 50 mM Tris-HCl buffer (pH 7.6; TB) (Sigma-Aldrich) and split into 1 mL aliquots. The samples were subjected to one of the following treatments: (i) 4 cycles of freezing and thawing; (ii) incubation with 2% IGEPAL^®^ CA-630 (NP-40) (Sigma-Aldrich) for 2 h at 37 °C, as per Qi et al.; (iii) sonication at 40 kHz for 4 min in ice water filtered through a 0.22 µm membrane (Merck, New Jersey, USA), and suspended in TB with 10% sodium deoxycholate (NaDOC/T) (Merck) and 1% EDTA trypsin for 2 h at 37 °C, as per Ideno et al.; or (iv) no treatment [24,25]. These are summarized in Table 2.

HEV-A patient stool filtrate and serum were used as controls, representing nHEV and eHEV, respectively. We loaded 1 mL of each sample into an ultracentrifugation tube, and the samples passed through an iodixanol gradient (2 mL each of 60%, 50%, 40%, 30%, 20%, and 10%) (Abbot, Chicago, IL, USA). The tubes were centrifuged at 160,000× *g* at 4 °C for 16 h. Twelve fractions of 1 mL were recovered, and the density and HEV RNA were measured for each fraction via refractometry and qRT-PCR, respectively.

### 2.8. Ribavirin Inhibition Assay

Cells were seeded and infected with HEV-A4 as described above; however, MEM-A at 0 dpi was supplemented with varying concentrations of ribavirin (200, 100, 75, 50 µg/mL). At 7 and 14 dpi, 50% of the supernatant was harvested and replaced with fresh supplemented media. Viral load in supernatant was then quantified as described above.

### 2.9. Statistical Analysis

Charts were generated using GraphPad Prism Version 8.1 (GraphPad software, La Jolla, CA, USA). Student’s *t* test and one-way ANOVA were used to compare mean viral loads in different media.

## 3. Results

### 3.1. Assessing the Effects of Hyper-Confluence, Amphotericin B, and MgCl_2_ on Culture Yield of HEV-A4 and HEV-C1

Schemmerer et al. identified hyper-confluent cell layers, amphotericin B, and MgCl_2_ as key factors that improved HEV-A genotype 3 (HEV-A3) yield from clinical samples in PLC/PRF/5 cells [19]. We attempted to replicate their findings by comparing HEV-A4 and HEV-C1 growth in PLC/PRF/5 cells with three different media: MEM-1, MEM-10, and MEM-A (Table 1). As shown in Figure 1a, MEM-10 boosted HEV-A4 yields only after 28 days post-inoculation (dpi) compared to MEM-1, likely because the former maintained cell layer integrity for a longer period of time.

Supplementing MEM-10 with amphotericin B and MgCl_2_ (MEM-A) further improved HEV-A4 yields. There was significantly higher growth of HEV-A4 in MEM-A (mean viral load: 3 × 10^5^ copies/mL) compared to MEM-1 (mean viral load: 4 × 10^2^ copies/mL with *p* = 0.017505). HEV-C1 RNA yield in PLC/PRF/5 cell supernatant was poor, with steadily decreasing supernatant virus loads across all media. HEV-A4-infected cells in MEM-A could be maintained for a period of 18 months with steady virus production over this time, with a maximum viral load of 3 × 10^8^ copies/mL (374 dpi) and an average viral load of 6 × 10^5^ copies/mL (Figure 2a). HEV-C1 infection could be maintained at low levels up to 3 weeks before a decline in viral load (Figure 2b). 

In order to confirm protein expression in our cell culture system, we performed an immunofluorescence assay as well as an ELISA. Immunofluorescence analysis at 7 dpi showed the presence of ORF2 antigen in cytoplasm of cells in both HEV-A4 and HEV-C1 infected cells (approximately 10–15% of cells were infected) (Figure 3). For the ELISA, we tested three supernatant samples representing 10^6^, 10^5^, and 10^4^ log_10_ copies/mL (corresponding to 91, 31, and 84 dpi, respectively, in Figure 2a). They were tested for HEV-Ag (ORF2) using a commercial ELISA. This resulted in optical density values of 2.71, 1.97, and 0.84, respectively, which exceeded the assay cutoff (corresponding to >0.25 WHO units/mL) and correlated with the respective viral loads (Figure 2a). We have shown that the Wantai kit cannot detect HEV-C1 [6]. Therefore, we could not replicate the confirmation of viral antigen in HEV-C1 supernatant.

### 3.2. Effects of Progesterone and DMSO on Culture Yield

We then examined the effects of adding progesterone to MEM-A (MEM-P). Recently, it has been demonstrated that supplementing media with progesterone significantly boosted HEV yield in an HEV-A3 infectious clone model (Kernow P6) in Huh7-S10-3 human liver cells [18]. We attempted to replicate these findings using patient-derived HEV-A4 and HEV-C1 at the same concentration of progesterone, 80 nM, recommended in this study protocol [18]. MEM-P did not significantly increase yields of either variant compared to MEM-A alone (Figure 4).

Next, we verified the findings that the presence of 1% DMSO in media promoted the growth of clinical strains of HEV-A in cell culture [17]. Growth of HEV-A was similar in MEM-D compared to MEM-A, although growth at 28 dpi tended to be higher in MEM-D than MEM-A, but this did not reach statistical significance. Growth of HEV-C1 was boosted by MEM-D (mean viral load: 3 × 10^4^ copies/mL) at 28 dpi compared to MEM-A (mean viral load: 9 × 10^2^ copies/mL) (*p* = 0.004785).

Finally, we hypothesized whether a medium combining amphotericin B, MgCl_2_, 1% DMSO, and 80 nM progesterone (MEM-Z) could boost viral growth. For HEV-A4, MEM-Z showed an increase after 28 dpi, growing to 7.70 × 10^7^ copies/mL compared to MEM-1, -10, -A, -P, and -D, which only reached a maximum of 4.01 × 10^5^ copies/mL (Figure 5a).

MEM-Z also increased the viral yield of HEV-C1 at 28 dpi, but to a lower, statistically insignificant level than MEM-D. This is consistent with the previous experiment finding that DMSO boosted growth of this variant, at least transiently (Figure 5b).

We also examined the effects of a range of concentrations of ribavirin on the growth of an HEV-A4 isolate. As shown in Figure 6, ribavirin had no discernible effect on viral replication kinetics of this isolate. This was expected because the patient from which this isolate was obtained had ribavirin refractory hepatitis E. 

### 3.3. Comparison of Methods for Converting eHEV to nHEV

When released from cells in vitro and in vivo, HEV is associated with a host cell membrane-derived envelope (eHEV). However, it loses this envelope when shed in feces due to the action of bile (nHEV) [26]. Yin et al. gave evidence of lower infectiousness of eHEV compared to nHEV, and attempts have been made to de-envelope the virions from cell culture supernatant through treatment with NP-40 and NaDOC/T [13,25,27,28,29]. Here, we compare the efficacy of various methods of converting eHEV to nHEV.

We first demonstrated that the density of HEV-A4 shed in cell culture supernatant (MEM-A) aligns with that of patient serum, confirming that cell-culture derived HEV is pseudo-enveloped (Figure 7a). After combining cell culture supernatant from multiple time points into a stock, we compared several treatments (Table 2) to identify the most effective method for shifting the density of virus particles towards that expected of nHEV (i.e., that found in stool). Incubation with 2% NP-40 was the most successful at de-enveloping eHEV particles, while four cycles of rapid freeze–thawing substantially reduced viral load and were relatively ineffective at converting eHEV to nHEV. Although there was a shift towards non-enveloped particles after treatment with NaDOC/T, there was a decrease in viral load compared to the NP-40 treatment (Figure 7b). The outcome of the NaDOC/T procedure was similar to that observed by Ideno et al. [25].

This experiment could not be repeated with HEV-C1 due to low viral yields.

## 4. Discussion

HEV is notoriously difficult to grow consistently in cell culture. Its growth is highly strain-, variant-, and cell line-dependent. Here, we attempted to replicate findings from three recent studies that proposed different cell culture modifications to boost HEV growth in vitro [17,18,19]. These modifications included hyper-confluent monolayers, amphotericin B, MgCl_2_, DMSO, and progesterone. Importantly, we standardized the evaluation by using a single cell line (PLC/PRF/5) and evaluated HEV variants that were not included in the original studies. Thus, we aimed to examine the generalizability of these previous studies.

Schemmerer et al. highlighted the importance of hyper-confluent cells (generated by growing them in 10% FBS for 7–14 days prior to infection) [19]. Both Schemmerer et al. and Berto et al. found higher susceptibility to HEV infections under such conditions, presumably due to closer contact between cells [19,30]. They speculated that the “3D matrix” better mimics the polarized nature of hepatocytes during a natural infection. Interestingly, we found little difference in HEV-A4 growth between hyper-confluent cells grown in MEM-1 or MEM-10, although only the latter supported viral yield beyond 28 dpi, likely because there were more adherent cells after prolonged incubation. Another key modification assessed was the addition of amphotericin B and MgCl_2_ to media. Media containing these components were clearly superior to MEM-10 for boosting HEV-A4 growth, confirming the findings of Schemmerer et al. [19]. Amphotericin B is known to improve virus replication in influenza cell culture, where it increases the acidification of internal cellular compartments and thus promotes a faster drop in pH within endosomes [31]. In the context of HEV, which enters hepatocytes via clathrin-mediated endocytosis that involves endosomal acidification, amphotericin B would be expected to have a similar beneficial effect [32,33]. MgCl_2_ is thought to shield virus particles from inactivating factors and thus increase viral titers [34].

DMSO has been postulated to increase the differentiation of hepatocytes and was noted by Capelli et al. to boost the growth of clinical HEV strains from genotypes 1 and 3 in HepG2/C3A cells or the HepG2/C3A subclone F2 [17]. The addition of DMSO appeared to have a transient effect on the yield of both HEV-A4 and HEV-C1 at certain time points. Sooriyanarain et al. found that progesterone was capable of improving infection efficiency of Huh7-S10-3 liver cells by HEV infectious clones [18]. This is an active area of investigation, given the increased hepatitis E severity in pregnancy. However, we could not replicate these findings in our model. We did not investigate the expression of progesterone receptors (PGRMC1) in PLC/PRF/5, which may have impacted the susceptibility of these cells to progesterone treatment. However, PLC/PRF/5 cells have been shown to express progesterone receptor membrane complex 1 (PGRMC1), which is involved in the non-classical progesterone binding pathway [35]. Our findings indicate that culture modifications must be independently evaluated using different cell lines. Our findings also illustrate that the behavior of wild-type viruses can be quite different from that of infectious clones. 

Overall, HEV-C1′s growth was weaker than HEV-A4 across all media. There is considerable genetic divergence between these two species. The HEV-C1 isolate used in this study only shares 56.5% of its nucleotide identity with HEV-A4 [9]. Clinically, HEV-C1 patients usually exhibit milder hepatitis than those infected with HEV-A [6,11]. This may indicate that HEV-C1 is less adapted to human hepatocytes than HEV-A4. Even so, we found that DMSO is a useful additive for improving HEV-C1 yields in human hepatoma cell lines.

Yin et al. explained that initial cell attachment is a major limiting factor in cellular infection; therefore, the presence of a quasi-envelope will impact access to cells during the infection process [13]. Several methods for converting eHEV to nHEV have been described in the literature. The most basic method is repeated freeze–thawing. Freeze–thawing is a procedure that results in the lysis of infected cells and membranes surrounding the virus particle, thus freeing them and increasing infectivity. However, our findings indicate that freeze–thawing only resulted in a slight shift in density, and also reduced viral loads, which would be expected to reduce yields. Chemical treatments such as NP-40 and NaDOC/T are designed to remove the lipid envelope. Both followed a similar density trajectory in Figure 7b, indicating success in de-enveloping the majority of HEV particles in supernatant. We speculate that the small difference in HEV RNA between the NP-40 and NaDOC/T treatments was due to the filtration step included in the NaDOC/T treatment protocol by Ideno et al. [24,25].

There are several limitations to this study. We used only one cell line, but PLC/PRF/5 is generally proven to be capable of supporting the growth of diverse HEV variants [19,22,25,26,27,29,36]. We also wanted to investigate whether the findings of previous studies were only applicable to certain genotypes and/or cell lines. Secondly, we lacked a conventional measure of viral yield, such as a TCID_50_. However, HEV does not produce cytopathic effects in cell culture, and TCID_50_ measurements based on genome copies do not necessarily reflect infectious virus. The presence of ORF2 antigen in the supernatant and success in passaging the virus supports the fact that cellular infection was productive, with mature viable virus particles in the supernatant. Our immunofluorescence assay data also show protein expression in cells. A recent pivotal study has found that that a combination of transfection of HepG2 cells by HEV-A3 infectious clone derived in vitro transcripts supplemented with complete DMEM, and subsequent infection of HepG2/C3A in MEM low IgG FCS yielded the highest intracellular-derived viral titers [37]. However, we were unable to apply this finding, as we did not have access to HEV-A4 and HEV-C1 infectious clone models. 

In summary, we have verified recently published protocols that DMSO, amphotericin B, and MgCl_2_ significantly improve HEV yield in vitro. We also systematically compared three methods to convert eHEV to nHEV, finding that NP40 is the most efficient. Our work lays a foundation for future efforts to optimize HEV cell culture systems. We would like to emphasize the importance of using wild-type virus derived from clinical specimens and different genotypes in such evaluations as much as possible.

## Figures and Tables

**Figure 1 viruses-14-01254-f001:**
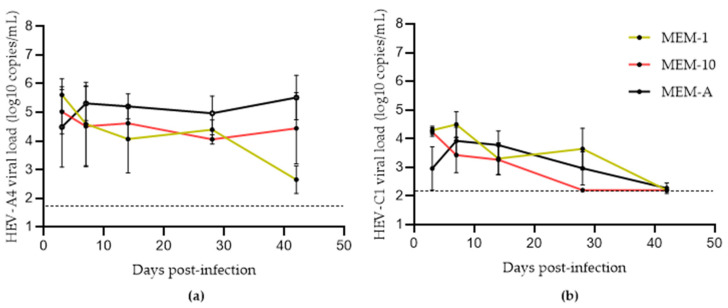
Growth of hepatitis E virus (HEV) in PLC/PRF/5 cells maintained in MEM-1, MEM-10, or MEM-A for (**a**) HEV-A genotype 4 (HEV-A4) and (**b**) HEV-C genotype 1 (HEV-C1) variants. Each data point represents the mean of three replicates. Error bars represent standard error of mean. Limit of detection of the RT-qPCR assay is represented by the dashed line (50 and 160 copies per mL for HEV-A4 and HEV-C1, respectively).

**Figure 2 viruses-14-01254-f002:**
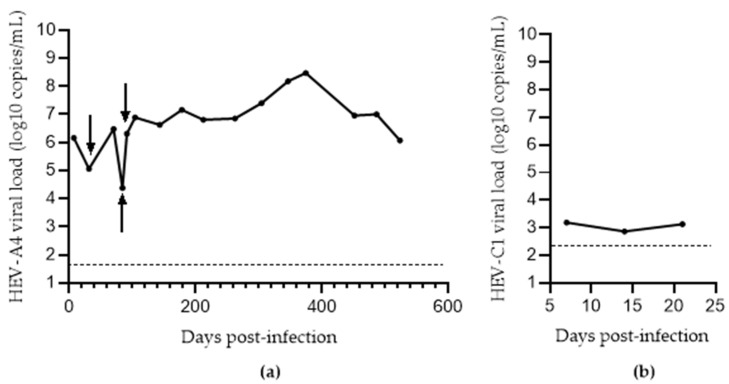
Growth of hepatitis E virus (HEV) in PLC/PRF/5 cells maintained in MEM-A for (**a**) HEV-A4 and (**b**) HEV-C1 variants. Limit of detection of the RT-qPCR assay is represented by the dashed line (50 and 160 copies per mL for HEV-A4 and HEV-C1, respectively). Arrows represent the points at which supernatant was tested for ORF2 antigen.

**Figure 3 viruses-14-01254-f003:**
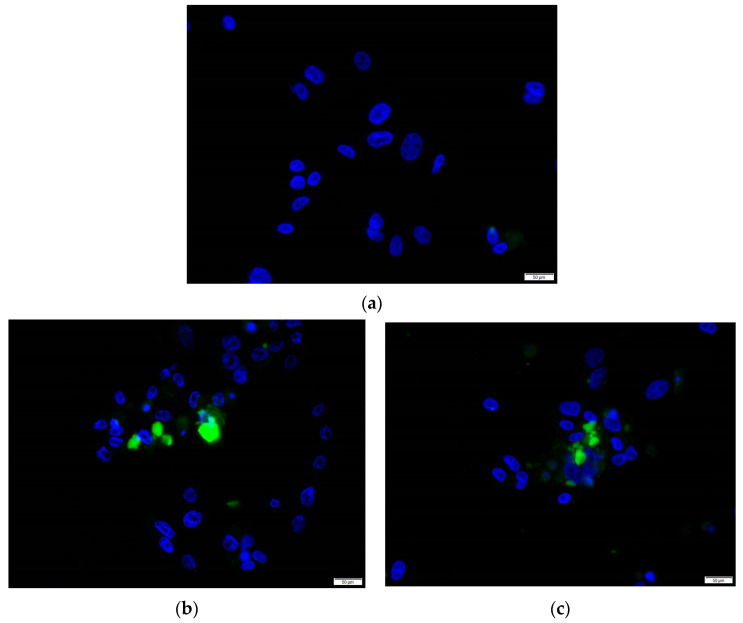
Immunofluorescence antibody staining of (**a**) uninfected PLC/PRF/5 cells, (**b**) HEV-A genotype 4 (HEV-A4)-infected cells, and (**c**) HEV-C genotype 1 (HEV-C1)-infected cells. Nuclei were counterstained with DAPI. Furthermore, we used HEV-A4 cell culture supernatant stock (1 × 10^5^ copies/mL) to infect naïve PLC/PRF/5 cells and maintained them for over 42 dpi at an average viral load of 5 × 10^4^ copies/mL, confirming that there was viable virus released into supernatant (Appendix A). This confirms that PLC/PRF/5 cells were productively infected.

**Figure 4 viruses-14-01254-f004:**
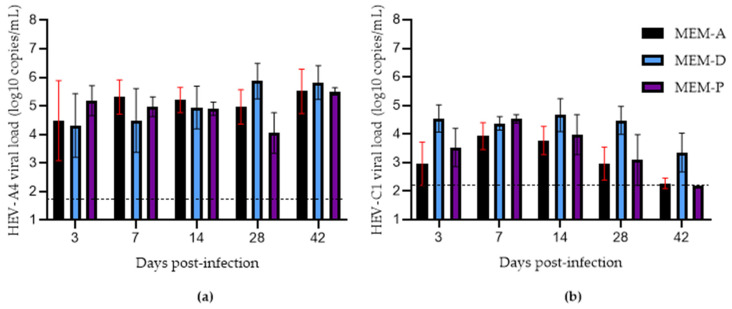
Growth of hepatitis E virus (HEV) in PLC/PRF/5 cells maintained in MEM-A and MEM-D and MEM-P for (**a**) HEV-A genotype 4 (HEV-A4) and (**b**) HEV-C genotype 1 (HEV-C1) variants. Each data point represents the mean of 3 replicates. Error bars represent standard error of mean. Limit of detection of the RT-qPCR assay is represented by the dashed line (50 and 160 copies per mL for HEV-A4 and HEV-C1, respectively).

**Figure 5 viruses-14-01254-f005:**
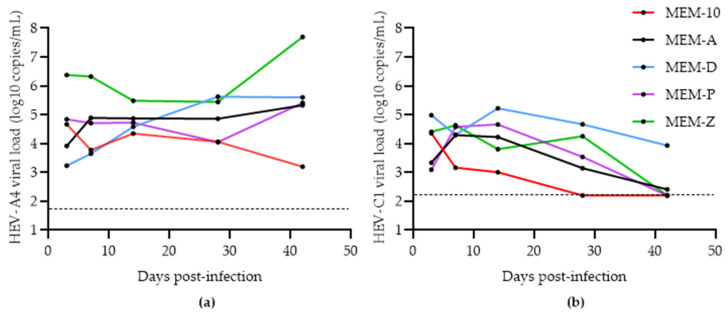
Growth of hepatitis E virus (HEV) in PLC/PRF/5 cells maintained in MEM-10, MEM-A, MEM-D, MEM-P, and MEM-Z for (**a**) HEV-A genotype 4 (HEV-A4) and (**b**) HEV-C genotype 1 (HEV-C1) variants. Each data point represents the mean of two replicates. Limit of detection of the RT-qPCR assay is represented by the dashed line (50 and 160 copies per mL for HEV-A4 and HEV-C1, respectively).

**Figure 6 viruses-14-01254-f006:**
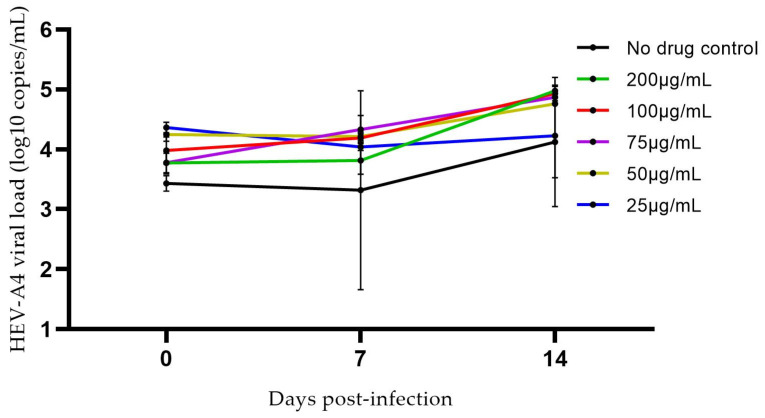
Growth of hepatitis E virus (HEV) in PLC/PRF/5 cells maintained in MEM-A supplemented with varying concentrations of ribavirin for HEV-A genotype 4 (HEV-A4). Each data point represents the mean of 3 replicates. Error bars represent standard error of mean. Limit of detection of the RT-qPCR assay is represented by the dashed line (50 copies per mL for HEV-A4).

**Figure 7 viruses-14-01254-f007:**
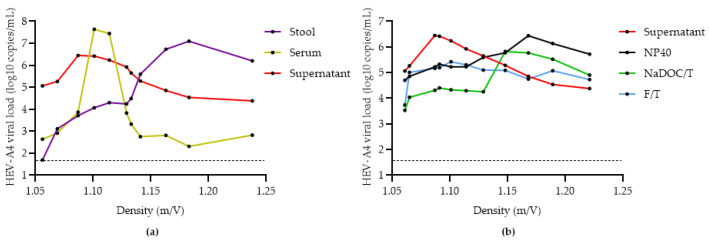
Ultracentrifuged iodixanol density gradient of (**a**) stool filtrate, serum, and untreated cell culture supernatant infected with HEV-A genotype 4 (HEV-A4); (**b**) HEV-A4-infected cell culture supernatant treated with NP-40, NaDOC/T, freeze–thaw cycle, and no treatment. Limit of detection of the RT-qPCR assay is represented by the dashed line (50 copies per mL for HEV-A4).

**Table 1 viruses-14-01254-t001:** Media composition.

Name	Composition
MEM-1	Minimum Essential Medium (MEM) supplemented with 1% heat-inactivated fetal bovine serum (FBS)
MEM-10	MEM supplemented with 10% FBS
MEM-A	MEM-10 supplemented with 2.5 µg/mL amphotericin B and 30 mM MgCl_2_
MEM-D	MEM-A supplemented with 1% dimethyl sulfoxide (DMSO)
MEM-P	MEM-A supplemented with 80 nM progesterone
MEM-Z	MEM-D supplemented with 80 nM progesterone

**Table 2 viruses-14-01254-t002:** Treatments of cell culture supernatant for iodixanol gradient experiment.

Name	Treatment
No treatment	No treatment as a control for quasi-enveloped cell culture-derived HEV
F-T	Repeated freeze–thaw cycle (4 times)
NP-40	2% NP-40 in PBS for 2 h at 37 °C
NaDOC/T	Sonicated twice at 40 kHz for 4 min in ice water, filtered through a 0.22 µm membrane, and suspended in TB with 10% sodium deoxycholate (NaDOC/T) and 1% EDTA trypsin for 2 h at 37 °C

## Data Availability

Not applicable.

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
