# Peer review of "Independent Evaluation of Cell Culture Systems for Hepatitis E Virus"

_viruses, 2022, doi:10.3390/v14061254_

Round 1

Reviewer 1 Report

Chew et al evaluated  the cell culture system of HEV. Using stool derived, HEV-4 and rat HEV, and different cell culture conditions.  The authors found that amphotericin B, MgCl2 and DMSO increased HEV yield, while progesterone not. HEV-A4 could be maintained for over 18 months in amphotericin B and MgCl2 containing medium with demonstration of 
viral antigen and viable virus in supernatant

In general, more experiments need to be done to confirm the author conclusions 

Major points

1- Assessment of intracellular HEV Ag/ protein is essential  in Figure 1 and Figure 2. The authors can not rely on HEV Ag assay which assessed extracellular Ag.  As ref 24, this assay can give false positive results and give results of different HEV ORF2 Ag. 

2- The protocol of infection is not clear. What is moi? how many cells used in the infections? Detailed protocols should be described

3- Importantly, the authors should provide the viral load in SN after virus removal (Day 0).  Washing and removal of inoculum can not remove all the virus and the residual can remain for longer period of time especially if the inoculums are stool derived. To study the kinetics, it is important to see the initial viral load in all experiments.

4- References: The authors cited 33 references, and 8 references are self citation, please try to reduce the self citation.

Reviewer 2 Report

In the here presented manuscript Chew and colleagues the authors evaluation the effectiveness of hyperconfluent cell layers, amphotericin B, MgCl2, progesterone, and DMSO modifications in improving the yield of HEV-A genotype 4 (HEV-A4) and HEV-C1 from clinical samples in PLC/PRF/5 cells. They find that yield of HEV-C1 was lower than HEV-A4 across all medium conditions, but was boosted by DMSO. Amphotericin B, MgCl2 and DMSO increased HEV-A4 yield from high viral load patient stool samples while progesterone was not effective. NP40 was further used to clear viral suspensions from enveloped particles.

This study is important as it underlines the problems with handling infectious HEV in cell culture. Whiel for HEV-A3 a robust tissue system was introduced in 2020, similar effective models are lacking for HEV-A1, A4 and C1.

General comments

The manuscript is clear and well-structured. It is scientifically sounded as the authors aimed at understanding. Although the study has some limitations (fully exposed by the authors at the end of the discussion), it remains that this study brings novelty and confirmation to studies that have been reported before. The cited literature is appropriate, also three suggestions are made below and ethical statement is adequate.

Specific comments

  • In my opinion, insertions in the HVR of HEV are no cell culture artifacts, as several insertions have been identified in patients (10.1038/s41598-022-05706-w, 10.3389/fmicb.2020.00001). By the way RSP17 in Kernow-C1 was also initially identified in a stool sample from a chronically infected patient.
  • Recently, there has been a study in PNAS (doi: 10.1073/pnas.1912307117) proposing an optimized system for HEV-A3 (e.g. line 59)
  • the authors should also show the course of HEV RNA under RBV treatment. This allows the reader to judge RNA in regard to replication specific effects
  • 162: the antigen ELISA does not score particle release, but only pORF2 presence.
  • MEM-Z seems to perform best for A4 while MEM-D for C1. Do the authors have any explanation for this? Please discuss or mention in more detail in the part l. 203
  • There is a control medium missing in Fig 3 without any modifications.
  • Line 167 and Figure S1: Please state at what time point the supernatant was harvested and used to test for viable virus.
  • In l. 166 I have the feeling Ag levels do not completely correlate with RNA load. RNA is medium, low, high, while pORF2 is high, medium, low. Do I interpret this correctly?
  • Did the authors also use a different cell line?
  • Is there another way of presenting the gradient data. To me it looks like there are no peaks at all, except for serum in the left panel. Can you discuss and compare to what is published?
  • Why are there different LOD for A4 and C1, even though according to reference 20, 5 µL template out of 200 µL sample were used for both RT-qPCRs? Do you mean LOQ?
  • Figure S1: dashed line corresponds to 160 copies/mL and not 50 copies/mL as referenced in the figure legend
  • Line 178, 194: “80 nM”
  • Line 214: Please state at what time point the supernatant was harvested
  •  

Round 2

Reviewer 1 Report

The authors addressed my questions. Only one minor point: the source of antibodies used in the IF should be mentioned.

Also approximately the percentage of infected cells.
